# ON LOGICAL EXTRAPOLATION FOR MAZES WITH RECURRENT AND IMPLICIT NETWORKS

## ABSTRACT

Recent work has suggested that certain neural network architectures—particularly recurrent neural networks (RNNs) and implicit neural networks (INNs)— are capable of *logical extrapolation*. That is, one may train such a network on easy instances of a specific task and then apply it successfully to more difficult instances of the same task. In this paper, we revisit this idea and show that (i) The capacity for extrapolation is less robust than previously suggested. Specifically, in the context of a maze-solving task, we show that while INNs (and some RNNs) are capable of generalizing to larger maze instances, they fail to generalize along axes of difficulty other than maze size. (ii) Models that are explicitly trained to converge to a fixed point (e.g. the INN we test) are likely to do so when extrapolating, while models that are not (e.g. the RNN we test) may exhibit more exotic limiting behaviour such as limit cycles, *even when* they correctly solve the problem. Our results suggest that (i) further study into *why* such networks extrapolate easily along certain axes of difficulty yet struggle with others is necessary, and (ii) analyzing the *dynamics* of extrapolation may yield insights into designing more efficient and interpretable logical extrapolators.

## 1 INTRODUCTION

A hallmark of human learning is the ability to generalize from easy problem instances to harder ones by merely thinking for longer. In (Schwarzschild et al., 2021b) it is demonstrated that recurrent neural networks (RNNs) are also capable of such *logical extrapolation*. That is, RNNs that are trained on 'easy' instances of a task such as solving small mazes can, in some cases, successfully solve 'harder' tasks of the same kind, such as larger mazes.

A pre-requisite for logical extrapolation is the ability of the network to adjust its computational budget to fit the difficulty of the problem at hand. Concretely, this means that the network should be able to vary its number of layers (or iterations). Two classes of network naturally fit this description: weight-tied RNNs and Implicit Neural

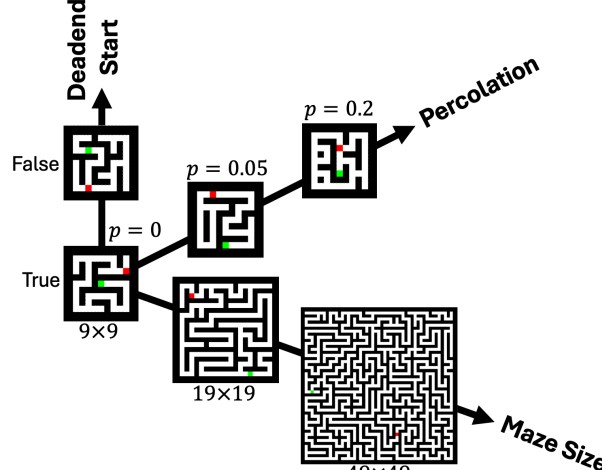

Figure 1: Three extrapolation dimensions: maze size, percolation, and deadend start. Each shown maze is generated using the indicated parameters. The origin (i.e., maze size $9 \times 9$, percolation $p = 0$, and `deadend_start = True`) represents the training distribution. Moving away from the origin corresponds to an out-of-distribution shift. Green denotes the start position.

Networks (INNs), also known as Deep Equilibrium Networks (DEQs) (Bai et al., 2019; El Ghaoui et al., 2021; Wu Fung et al., 2022). Both INNs and RNNs, described further in Section 2, have been considered for logical extrapolation problems (Schwarzschild et al., 2021b;c; Bansal et al., 2022; Anil et al., 2022) .

In this work we revisit prior results on logical extrapolation with both RNNs and INNs in the context of a single task: maze-solving (Bansal et al., 2022; Anil et al., 2022). While previous studies on logical extrapolation in maze-solving have characterized difficulty as simply a function of maze size, we introduce two new ways in which to vary the difficulty: (i) a binary variable `deadend_start` $\in$ {`True`, `False`} which, when set to `True`, constrains the start point to have exactly degree 1, (ii) a percolation constant $p \in [0, 1]$ which relates to the likelihood of a maze containing loops. Only for $p = 0$ are mazes are guaranteed to be acyclic and thus have unique solutions. We note that in both prior works examining logical extrapolation in maze-solving, `deadend_start = True` and $p = 0$. We show that the models introduced in prior work (Bansal et al., 2022; Anil et al., 2022) *do not generalize* when `deadend_start` and $p$ are varied, even though they generalize well as maze size is varied. More detail on the maze solving task and the modifications we make is given in Section 3.2.

As our second major contribution, we investigate *how* RNNs and INNs generalize. Initially, it was observed that with proper training RNNs converge to a fixed point, even for more difficult task instances such as larger maze sizes (Bansal et al., 2022). However, Anil et al. (2022) hints at more complex behaviour, as they find evidence of periodicity in the dynamics of INNs when applied to larger mazes. We quantify this phenomenon using tools from *Topological Data Analysis* (TDA)(De Silva et al., 2012; Perea & Harer, 2015; Tralie & Perea, 2018). We find that, while the INN we consider (Anil et al., 2022) consistently converges to a fixed point, regardless of maze size, the RNN we consider (Bansal et al., 2022) exhibits more complex limiting behaviour. Specifically, for most larger mazes, this RNN converges to either a two-point cycle or two-loop cycle. In order to streamline TDA on INN/RNN architectures, this work also contributes a `PyTorch`-based (Paszke et al., 2019) wrapper to `Ripser` (Bauer, 2021; Tralie et al., 2018), a fast Python library for TDA (see Subsection 2.3 and Appendix D for more details).

Our results suggest that a network's ability to extrapolate may depend on the axis along which difficulty is increased; thus, greater caution is needed when using neural networks for extrapolation. We conclude by discussing how the tools introduced for studying periodicity may also be useful in other deep learning contexts.

## 2 BACKGROUND AND PRIOR WORK

### 2.1 RECURRENT NEURAL NETWORKS

A special class of RNNs, namely, weight-tied input-injected networks (or simply weight-tied RNNs), are used in logical extrapolation (Schwarzschild et al., 2021b; Bansal et al., 2022). For a $K$-layer weight-tied RNN $\mathcal{N}_\Theta$, the output is given by

$$\mathcal{N}_\Theta(d) = P_{\Theta_2}(u_K) \text{ where } u_j = T_{\Theta_1}(u_{j-1}, d) \text{ for } j = 1, \ldots, K. \tag{1}$$

Here, $\Theta_1$ and $\Theta_2$ are the parameters of the networks $T_{\Theta_1}$ and $P_{\Theta_2}$ respectively, and $\Theta := \{\Theta_1, \Theta_2\}$, while $d$ denotes the input features. These networks represent a unique class of architectures that leverage weight sharing across layers to reduce the number of parameters. The input injection at each layer ensures the network does not 'forget' the initial data (Bansal et al., 2022). In (Bansal et al., 2022), it is empirically observed that a certain weight-tied RNN extrapolates to larger mazes when applying more iterations. The authors speculate that the reason for this success is that the model has learned to converge to fixed points within its latent space (Section 5, Bansal et al. (2022)).

### 2.2 IMPLICIT NETWORKS

Drawing motivation from (Bansal et al., 2022), (Anil et al., 2022) propose to use implicit neural networks (INNs) for logical extrapolation tasks. INNs are a broad class of architectures whose outputs are the fixed points of an operator parameterized by a neural network. That is,

$$\mathcal{N}_\Theta(d) = P_{\Theta_2}(u_\star) \quad \text{where} \quad u_\star = T_{\Theta_1}(u_\star, d). \tag{2}$$

Here again $\Theta = \{\Theta_1, \Theta_2\}$ refers collectively to the parameters of the networks $T_{\Theta_1}$ and $P_{\Theta_2}$ and $d$ is the input feature, while $u_\star$ represents a fixed point of $T_\Theta$. These networks can be interpreted as

infinite-depth weight-tied input-injected neural networks (El Ghaoui et al., 2021; Bai et al., 2019; Winston & Kolter, 2020).

Unlike traditional networks, INN outputs are not defined by a fixed number of computations but rather by an implicit condition. INNs have been applied to domains as diverse as image classification (Bai et al., 2020), inverse problems (Gilton et al., 2021; Yin et al., 2022; Liu et al., 2022; Heaton et al., 2021; Heaton & Wu Fung, 2023), optical flow estimation (Bai et al., 2022), game theory (McKenzie et al., 2024a), and decision-focused learning (McKenzie et al., 2024b). In principle, INNs are naturally suited for logical extrapolation, as they are not defined via an explicit cascade of layers, but rather by an implicit, fixed-point, condition. This condition can be viewed as specifying when the problem is considered solved. Key to logical extrapolation is that this characterization of "solving a problem" is always the same, regardless of the difficulty of the problem at hand.

### 2.3 TOPOLOGICAL DATA ANALYSIS IN THE LATENT SPACE

For both RNNs and INNs, we call $\{u_j\}_{j=1}^K \subset \mathbb{R}^n$ the *latent iterates*, and $n$ the latent dimension. Note that $n$ can be, and often is, larger than the dimension of the input feature $d$ or network output $\mathcal{N}_\Theta(d)$. To characterize the limiting behaviour of the sequence of latent iterates we study its *shape*. Intuitively, if this sequence exhibits periodic behaviour, it should trace out a loop. More precisely, the sequence should appear as though it was sampled from the topological equivalent of a circle embedded in $\mathbb{R}^n$. Topological data analysis (TDA) provides a set of tools for analysing the shape of point clouds, and has been previously applied in other contexts to study periodicity (De Silva et al., 2012; Perea & Harer, 2015; Tralie & Perea, 2018).

As stated in Section 1, we construct a `PyTorch` wrapper to `Ripser` (Bauer, 2021; Tralie et al., 2018), a fast Python library for TDA (see Appendix D for further details). `Ripser` computes (persistent) homology groups to identify the most significant topological features of the point cloud. The relevant quantities are the zeroth and first (persistent) Betti numbers, which we denote as $B_0$ and $B_1$ respectively. These are the "dimensions"[1] of the zeroth and first homology groups, and count the respective number of connected components and loops in the data. We identify and interpret three common values of the tuple $[B_0, B_1]$.

1. **Convergence to a point** ($[B_0, B_1] = [1, 0]$). The sequence is clustered around a single point. No loops are present.

2. **Two-point cycle** ($[B_0, B_1] = [2, 0]$) The sequence is clustered around two points, and alternates between them. No loops are present.

3. **Two-loop cycle** ($[B_0, B_1] = [2, 2]$) The sequence lies along two well-separated, thickened loops, and alternates between them.

We emphasize that, while 1. represents the expected convergent behaviour (see Sections 2.1, 2.2) 2. and 3. represent novel, previously undetected limiting behaviour. A more precise overview of TDA is presented in Appendix B.

## 3 EXPERIMENTS

We study two trained maze-solving models from previous works. The first model is an RNN from Bansal et al. (2022) which we call `DT-Net`, and the second is an INN from Anil et al. (2022) which we call `PI-Net`[2]. We emphasize that while both works propose multiple models, we focus on the most performant model from each work. Our source code is provided in the supplementary material, and will be released publicly.

`DT-Net` uses a progressive loss function to encourage improvements at each RNN layer. In this approach, the recurrent module is run for a random number of iterations, and the resulting output is used as the initial input for the RNN, while gradients from the initial iterations are discarded. The

---

[1] More formally: these count classes in the zeroth and first homology groups of the Rips complex that persist for large ranges of the length scale.

[2] This stands for 'Path-Independent' net, as path independence is a feature identified in Anil et al. (2022) as being strongly correlated with generalization.

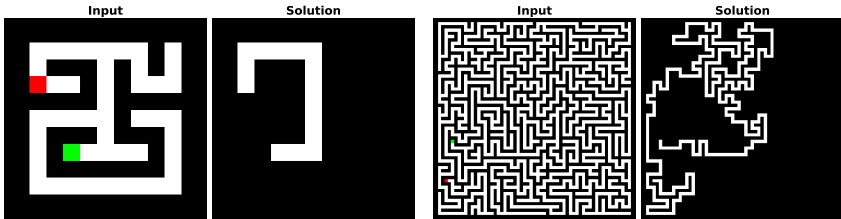

Figure 2: Maze input-solution pairs of size $9 \times 9$ (left) and $49 \times 49$ (right). Start positions are in green and end positions are in red. Mazes problems/inputs are RGB raster images and solutions are black and white images highlighting the solution path in white.

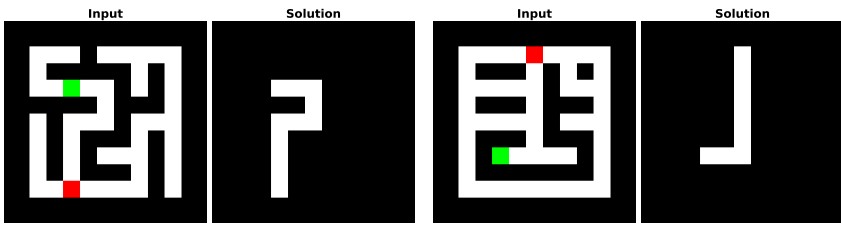

Figure 3: Example of maze with a start position that has multiple neighbors (left); example of a percolated maze ($p = 0.2$) with loops (right).

model is then trained to produce the solution after another random number of iterations. We refer the reader to (Bansal et al., 2022, Section 3.2) for additional details. For `PI-Net`, path-independence (i.e., contractivity) is encouraged in two ways: (i) by using random initialization for half of the batch and zero initialization for the other half, and (ii) by varying the compute budgets/depths of the forward solver during training.

### 3.1 THE MAZE-SOLVING TASK

In this and previous work (Schwarzschild et al., 2021a;b; Bansal et al., 2022; Anil et al., 2022), maze solving problems are encoded as raster images ( Figure 2). Given this RGB input image, the task is to return a black and white image representing the unique path from start (indicated by a green tile) to end (indicated by a red tile). In this work, we consider "accuracy" on a single maze to be 1 if the solution is exactly correct and 0 otherwise. However, instead of using the original "easy to hard" dataset (Schwarzschild et al., 2021a), we use the `maze-dataset` Python package (Ivanitskiy et al., 2023). `maze-dataset` can provide maze-solution pairs in the same format and from the same distribution, but allows modifications to the distribution if desired. The original "easy to hard" dataset only contains acyclic (i.e. percolation parameter $p = 0$, any path between two nodes is unique) mazes generated via randomized depth-first search (RDFS) and with start positions having exactly degree 1 (i.e., `deadend_start=True`). We remove these restrictions to investigate the behavior of the selected models on out-of-distribution mazes in Subsection 3.2. More details on our usage of `maze-dataset` are given in Appendix A.

### 3.2 EXTRAPOLATION

Usage of the `maze-dataset` package allows us to explore the behavior of `DT-Net` and `PI-Net` outside of the training distribution in a direction other than simply maze size. Specifically, in addition to being able to create mazes of any size (Figure 2), we investigate mazes whose start is not restricted to nodes of degree 1 (Figure 3). Furthermore, by setting the percolation parameter $p$ to values $> 0$, we can create mazes that may contain cycles, which means that the uniqueness of valid paths or even shortest-path solutions is no longer guaranteed (Figure 3). More detail on `maze-dataset` and our usage of it is given in Appendix A of the appendix and Subsection 3.1.

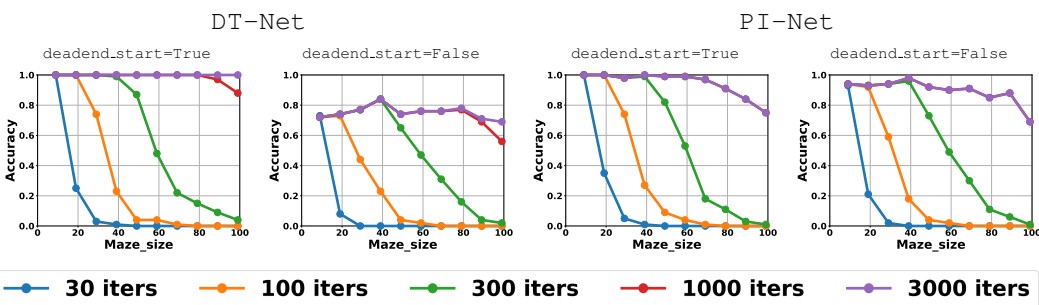

Figure 4: **Left:** `DT-Net` extrapolation accuracy (see Subsection 3.1) on a sample of 100 mazes at various maze sizes, with `deadend_start=True` and `deadend_start=False`. **Right:** Analogous results for `PI-Net`. Both models extrapolate very well to larger maze sizes with sufficient iterations. However, performance diminishes when the start position is allowed to have neighbors, regardless of the number of iterations. See Subsection C.3 for examples of these failures. Note that `deadend_start=False` does not guarantee that the degree of the start position is $> 1$, and mazes still satisfying this condition contribute to some of the performance seen. See Figure 3.2 for a breakdown of accuracy by start position.

**Increased Maze Size**. We first verify the extrapolation performance of `DT-Net` and `PI-Net` with increasing maze size (Bansal et al., 2022; Anil et al., 2022). For each maze size $n \times n$, where $n \in \{9, 19, 29, \ldots, 99\}$, we tested each model on 100 mazes. As expected, with sufficient iterations, both models achieve strong performance. See the plots labeled `deadend_start=True` in Subsection 3.2. Both models achieve perfect accuracy on the $9 \times 9$ mazes of the training distribution. Furthermore, with 3,000 iterations, `DT-Net` achieves perfect accuracy and correctly solved all test mazes. `PI-Net` achieved near perfect accuracy on smaller mazes, but performance noticeably diminished for mazes larger than $59 \times 59$[3]. Importantly, running more iterations usually helps and never harms accuracy. Note that for `PI-Net` the performance of the model after 1,000 iterations is identical to performance after 3,000 iterations at all tested maze sizes; this indicates that convergence occurred by 1,000 iterations.

**Deadend Start**. Allowing the start position to have multiple neighbors, rather than starting at a deadend, represents a different out-of-distribution shift from the training dataset. This shift corresponds to changing `deadend_start` from `True` to `False`, and diminishes the performance of both models. See the plots labeled `deadend_start=False` in Subsection 3.2. With this shift, accuracy on $9 \times 9$ mazes drops from 1.00 to 0.72 for `DT-Net` and from 1.00 to 0.94 for `PI-Net`. Interestingly, the fraction of failed predictions remains relatively stable as maze size is increased. There is no clear qualitative difference between mazes that were correctly and incorrectly solved by the models. However, we do observe that accuracy decreases monotonically from 1.0 as the number of start position neighbors increases from 1, a deadend, to 4, the maximum neighbors possible (see Figure 3.2). See Appendix C.3 for examples of mazes the models fail to solve.

**Percolation**. The final out-of-distribution shift we consider is increasing percolation from 0 in the training dataset, to a nonzero value which potentially introduces cycles into the mazes. We reiterate that when percolation equals 0, all mazes are acyclic and hence any path between two nodes is unique. This shift significantly reduces accuracy, as shown in Figure 3.2, and it highlights the ill-posed nature of solving percolated mazes. The presence of loops creates multiple paths to the goal. Notably, increasing the number of iterations in this setting *does not improve model performance*. We also observe that the loops introduced by percolation persist during inference with `DT-Net`, indicating behavior similar to that of the dead-end-filling algorithm (Hendrawan, 2020).

### 3.3 Latent Dynamics

For both `DT-net` and `PI-net` the latent space dimension $n$ is significantly larger than the output space dimension. Consequently, $P_{\Theta_2}$ is a projection operator with large-dimensional fibers[4]. While prior works emphasize the importance of training a model to reduce loss, i.e. the discrepancy be-

---

[3]Note this does not contradict the experiments in Anil et al. (2022)

[4]By *fiber* we are referring to the preimage of any point in the output space under $P_{\Theta_2}$.

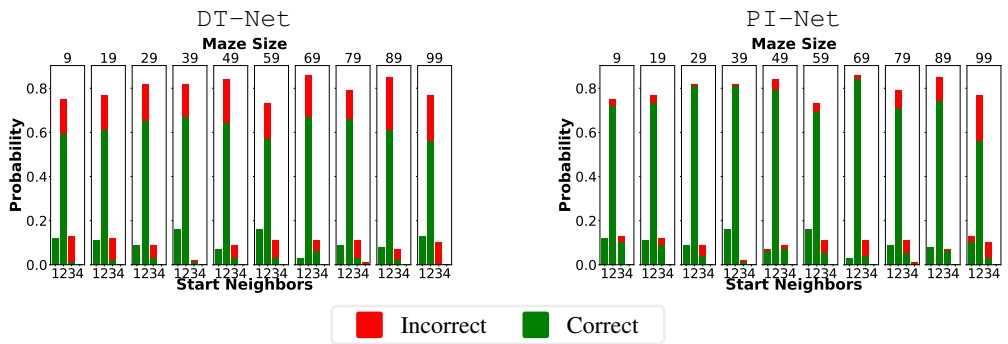

Figure 5: `DT-Net` and `PI-Net` predictions for `deadend_start=False` maze predictions split by maze size and the number of start position neighbors (from 1, a deadend, to 4, the maximum possible). For both models, accuracy diminishes on mazes with more start position neighbors.

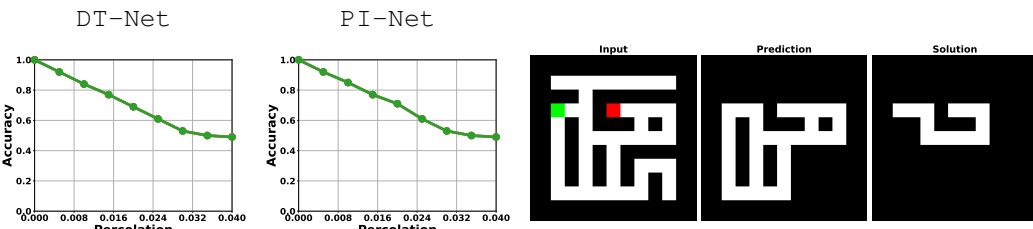

Figure 6: **Left:** Accuracy rapidly diminishes for both models when percolation increases above 0. Both models were iterated 30 times, and the resulting outputs do not change with additional iterations. **Right:** Both models fail on any maze with a loop, as they always include loops in the prediction.

tween $\mathcal{N}_\Theta(d)$ and the true solution $x^\star$, at every iteration (Bansal et al., 2022; Anil et al., 2022), there is no incentive for the iterative part of the network $T_{\Theta_1}(\cdot, d)$ to prefer one element of the fiber $P_{\Theta_2}^{-1}(x^\star) := \{u \in \mathbb{R}^n : P_{\Theta_2}(u) = x^\star\}$ over another. Thus, $T_{\Theta_1}(\cdot, d)$ may exhibit more complex dynamics than convergence-to-a-point, while $\mathcal{N}_\Theta(d)$ still yields the correct solution.

This possibility is considered for the first time in Anil et al. (2022), where it is proposed that, in order to solve a particular instance, $T_{\Theta_1}(\cdot, d)$ need not have a unique fixed point, but rather need only possess a global attractor. In other words, no matter which initialization $u_0$ is selected, the latent iterates exhibit the same asymptotic behaviour. They dub this property "path independence". In (Anil et al., 2022, App. F, App. G) evidence of instances $d$ where the latent iterates induced by $T_{\Theta_1}(\cdot, d)$ form a limit cycle, yet $\mathcal{N}_\Theta(d)$ is correct, is provided.

It is therefore both interesting and important to understand the latent dynamics of `DT-Net` and `PI-Net`. Building upon Anil et al. (2022), we introduce several tools for doing so. Most importantly, we use TDA 2.3 to *quantitatively* study the statistics of the limiting behaviours induced by a pretrained $T_{\Theta_1}(\cdot, d)$ as $d$ varies. In our experiments, for both models, we consider 100 mazes at maze sizes $9 \times 9$, $19 \times 19, \ldots, 69 \times 69$. We select a "burn-in" parameter $\tilde{K} < K$ and then consider latent iterates $\{u_j\}_{j=\tilde{K}}^K$ in order to study stable long-term latent behavior. We set $\tilde{K} = 3,001$ and $K = 3,400$.

**Residuals**. (Anil et al., 2022) considers the residuals $r_j := \|u_{j+1} - u_j\|_2$, i.e. the distances between consecutive iterates. The one-dimensional sequence of residuals offers a window into the high-dimensional dynamics of the latent iterates. In particular, if $r_j = 0$ for all sufficiently large $j$ then the $u_j$ have converged to a fixed point. (Anil et al., 2022) finds instances $d$ such that the residual sequence $\{r_j\}_{j=\tilde{K}}^K$ induced by a variant[5] of `PI-net` is visually periodic. We replicate this finding for `DT-net` (see Figure 7, third panel) and discover a novel asymptotic behaviour of

---

[5]Although not the variant we consider, see Appendix C.1 for further discussion.

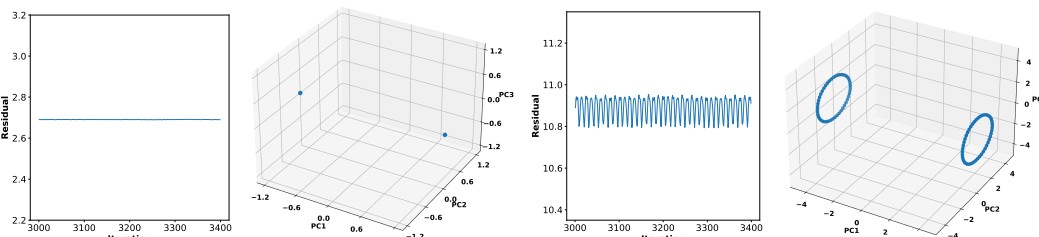

Figure 7: Residual plots and corresponding PCA projections for two sequences of `DT-Net` latent iterates. The left two plots indicate oscillation between two points corresponding to $[B_0, B_1] = [2, 0]$ for a $19 \times 19$ maze. The right two plots indicate oscillation between two loops corresponding to $[B_0, B_1] = [2, 2]$ for a $69 \times 69$ maze. Both mazes were solved correctly.

$\{r_j\}_{j=\tilde{K}}^{K}$: convergence to a *nonzero value* (see Figure 7, first panel). To understand the underlying latent dynamics more deeply, a different method is required.

**PCA**. Projecting the high-dimensional latent iterates onto their first three principal components reveals the underlying geometry of $\{u_j\}_{j=\tilde{K}}^{K}$ responsible for the observed residual sequences $\{r_j\}_{j=\tilde{K}}^{K}$. Specifically, the first sequence of latent iterates oscillates between two points (see Figure 7, panel 2), yielding constant values of $r_j$ equal to the distance between these points. We call such limiting behaviour a *two-point cycle*. The second sequence of latent iterates oscillates between two loops (see Figure 7, panel 4), yielding values of $r_j$ that oscillate around the distance between these loops. We call such limiting behaviour a *two-loop cycle*. To the best of our knowledge, neither of these limiting behaviours has been observed previously in the latent dynamics of an RNN or INN.

**TDA**. Using TDA tools as discussed in Section 2.3, we analyze the frequency with which the aforementioned limiting behaviours occur. This is possible because these limiting behaviours are distinguishable using the persistent Betti numbers (see Section 2.3 and Appendix B) of $\{u_j\}_{j=\tilde{K}}^{K}$. Specifically, convergence to a point has $[B_0, B_1] = [1, 0]$, a two-point cycle has ($[B_0, B_1] = [2, 0]$), and a two-loop cycle ($[B_0, B_1] = [2, 2]$).

Table 4 summarizes the TDA results. For `PI-Net`, every latent sequence converges to a fixed-point. For `DT-Net` at every maze size the majority of latent sequences approach a two-point cycle, a minority approach a two-loop cycle, and a few approach some other geometry. Interestingly, `DT-Net` exhibits fixed-point convergence in latent sequences of 17 in-distribution mazes at maze size $9 \times 9$.

## 4 DISCUSSION

The results of Subsection 3.2 suggest two points warranting further discussion. First, we highlight that seemingly mild distribution shifts (e.g. that induced by toggling `deadend_start`) can have a large negative effect on model performance, while performance can be unchanged under a seemingly larger distribution shift (e.g. increasing maze size). Secondly, other distribution shifts (e.g. using a nonzero percolation parameter and thus allowing for maze cycles) may make the task, as framed by the training data given to the model, ill-posed. More specifically, both `DT-net` and `PI-net` are trained to find the *unique* path from start to end, but when the maze has even a single loop, there is no longer a unique path. When presented with a maze that does not have a unique solution path, both models fail (see Figure 3.2 and Appendix Subsection C.3), whereas a human might reasonably reinterpret the task (e.g. "find the shortest path", or even "find a path") and solve it.

The results of Subsection 3.3 suggest that the dynamics of RNNs are richer than previously thought, particularly when the latent space is high-dimensional. While Section 4 clearly shows that models trained with path independence (Bansal et al., 2022) converge to fixed points more frequently, it is unclear how this correlates with (i) overall model accuracy and (ii) robustness towards distributional shifts. If allowing more exotic limiting behaviour (e.g. limit cycles, not just fixed points) is benign, or even desirable, various theoretical results on the convergence and backpropagation of RNNs and INNs (Liao et al., 2018; Wu Fung et al., 2022; Ramzi et al., 2022; Geng et al.; Bolte et al.,

Table 1: Betti number frequencies for `DT-Net` and `PI-Net`. `PI-Net` always exhibits fixed-point convergence ($[B_0, B_1] = [1, 0]$) whereas `DT-Net` usually approach a two-point cycle ($[B_0, B_1] = [2, 0]$) or sometimes a two-loop cycle ($[B_0, B_1] = [2, 2]$). *Sequence converged to within 0.01.

| MODEL | $[\mathbf{B_0}, \mathbf{B_1}]$ | \multicolumn{7}{c}{$n$ for $n \times n$ maze} | | | | | | |
| | | 9 | 19 | 29 | 39 | 49 | 59 | 69 |
|---|---|---|---|---|---|---|---|---|
| `DT-Net` | $[1, 0]^*$ | 17 | 0 | 0 | 0 | 0 | 0 | 0 |
| | $[2, 0]$ | 75 | 80 | 79 | 74 | 73 | 77 | 86 |
| | $[2, 2]$ | 4 | 18 | 17 | 22 | 25 | 21 | 14 |
| | Other | 4 | 2 | 4 | 4 | 2 | 2 | 0 |
| `PI-Net` | $[1, 0]^*$ | 100 | 100 | 100 | 100 | 100 | 100 | 100 |

2024) need to be revisited and adjusted to allow such behaviour. If exotic limiting behaviour is in fact undesirable, further research on interventions promoting path independence Winston & Kolter (2020); Bansal et al. (2022), and the ensuing tradeoffs, should be conducted.

The exploration of neural networks' internal representations in maze-solving scenarios has emerged as a compelling area of study (Mini et al., 2023; Ivanitskiy et al., 2024). This research aligns closely with the rapidly expanding field of AI interpretability (Räuker et al., 2023), which has become increasingly crucial as neural architectures grow in sophistication. Investigations into networks trained on spatial tasks—spanning chess (Karvonen, 2024; Jenner et al., 2024; McGrath et al., 2022), othello (Li et al., 2022; Nanda, 2023; He et al., 2024), graph traversal (Brinkmann et al., 2024; Momennejad et al., 2024), and mazes (Mini et al., 2023; Ivanitskiy et al., 2024)—have provided significant insights into their decision-making processes. Our study of the `DT-Net` model, with its foundation in chess puzzle training (Bansal et al., 2022; Schwarzschild et al., 2021b), contributes another valuable perspective to this complex landscape of spatial reasoning research.

Our topological tools could be used to explore how distribution shifts affect the latent dynamics, complementing prior work Liang et al. (2021) which considers this from a non-topological perspective. It would be useful to determine if topological information can be used to detect out-of-distribution examples, analogous to how Sastry & Oore (2020) flags examples with abnormal latent representations using Gram matrices. Finally, we note that the tools developed in this work could be applied to study latent dynamics in other settings, for example data assimilation (Williams et al., 2023).

**Limitations.** While this work focuses on the two most performant models from Bansal et al. (2022) and Anil et al. (2022), considering other models proposed in these works may yield additional insights. Moreover, it would be of interest to consider different dimensions of extrapolation for other tasks considered in the aforementioned works, for example the prefix sum problem or solving chess puzzles Schwarzschild et al. (2021a). We did not do so as it is less clear (to us) how to define, and interpret, such dimensions.

## 5 CONCLUSION

Using a maze-solving task with out-of-distribution test datasets constructed along different axes (maze size, deadend start, and percolation), we demonstrate that the ability of RNNs or INNs to extrapolate can depend on the type of out-of-distribution shift considered. Specifically, we find that a trained RNN and INN (`DT-Net` and `PI-Net`, respectively), can successfully extrapolate maze-solving to larger mazes but are less successful in extrapolating maze-solving to mazes with start positions with multiple neighbors and mazes with loops.

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

# A   ADDITIONAL MAZE DATASET DETAILS

Both `DT-Net` and `PI-Net` were trained on the same maze dataset (Schwarzschild et al., 2021a) containing $50,000$ mazes of size $9 \times 9$, meaning the mazes are subgraphs of the $5 \times 5$ lattice. Training mazes were generated via RDFS without percolation, and with the start position being constrained to being at a dead end (exactly 1 neighbor). We mimic this training distribution and add out-of-distribution shifts using the `maze-dataset` Python package (Ivanitskiy et al., 2023).

We first note that for controlling maze size, `maze-dataset.MazeDatasetConfig` takes a parameter `grid_n` which denotes the size of the lattice which the maze is a subgraph of. By contrast, the "easy to hard" (Schwarzschild et al., 2021a) dataset considers an $n \times n$ mean that many blocks in its raster representation. To convert between these two notions of maze size, $n = 2 \cdot (\texttt{grid\_n}) - 1$.

We can modify the start position constraint in `maze-dataset` by setting `deadend_start=False` in `endpoint_kwargs`. When `False`, the start position is sampled uniformly at random from all valid nodes, while when `True` the start position is samples uniformly at random from all valid nodes with degree one. Valid nodes are, by default, those not directly matching the end position or directly adjacent to it.

Randomized depth-first search (RDFS) is a standard algorithm for generating mazes, and produces mazes which are spanning trees of the underlying lattice, and thus do not contain cycles. For any acyclic graph with a spanning connected component, there is a unique (non-backtracking) path between any pair of points, and thus solutions are guaranteed to be unique. In our work, we select `LatticeMazeGenerators.gen_dfs_percolation` as the `maze_ctor` parameter. This function takes an additional variable `p` in `maze_ctor_kwargs`, which controls the percolation parameter. This percolation parameter, denoted $p$ in our work, means that the final maze is the result of a logical `OR` operation on the presence of all possible edges in the maze between an initial maze generated via RDFS and a maze generated via percolation, where each edge is set to exist with probability $p$. This is equivalent to first generating a maze with RDFS and then setting each wall to an edge with proability $p$. Since the initial RDFS maze is a spanning tree, adding any edge will cause the creation of a cycle, thus giving our desired out-of-distribution mazes.

# B   TOPOLOGICAL DATA ANALYSIS

Topological Data Analysis, or TDA, attempts to produce informative summaries of high dimensional data, typically thought of as point clouds[6] $\mathcal{U} = \{u_1, \ldots, u_K\} \subset \mathbb{R}^n$, by adapting tools from algebraic topology.

## B.1   SIMPLICIAL COMPLEXES

We are interested in TDA tools based on the idea of homology groups (Hatcher, 2002, Chapter 2). Homology groups can be computed algorithmically from a geometric object known as a *simplicial complex*, which we define below.

**Definition 1.** We collect definitions of several relevant concepts related to simplices.

1. A $k$-simplex is the convex hull of any $k + 1$ points in $\mathbb{R}^k$,
$$\sigma := \text{Conv}\{u_1, \ldots, u_{k+1}\} \tag{3}$$
$$= \left\{ \sum_{i=1}^{k+1} \alpha_i u_i : \ \alpha_i \geq 0 \text{ and } \sum_{i=1}^{k+1} \alpha_i = 1 \right\}$$

2. A face of a $k$-simplex $\sigma$ is a piece of the boundary of $\sigma$ which is itself a simplex. That is, $\tau$ is a face of $\sigma$ defined in equation 3 if
$$\tau = \text{Conv}\left\{u_{i_1}, \ldots, u_{i_{\ell+1}}\right\}$$

3. A simplicial complex $\mathcal{S}$ is a set of simplices, of multiple dimensions, satisfying the following properties

---

[6]By using the term "point cloud" we are implying that the ordering of points does not matter.

(a) For all $\sigma \in \mathcal{S}$, all faces of $\sigma$ are also in $\mathcal{S}$.

(b) If any two $\sigma_1, \sigma_2 \in \mathcal{S}$ have non-empty intersection, then $\sigma_1 \cap \sigma_2$ is a face of both $\sigma_1$ and $\sigma_2$, and consequently $\sigma_1 \cap \sigma_2 \in \mathcal{S}$

The process of computing homology groups from a simplicial complex is beyond the scope of this paper. We refer the reader to (Hatcher, 2002, Chapter 2) for further details. It is also possible to define homology groups for *abstract simplicial complexes*, $\mathcal{R}$, for which a $k$-simplex $\sigma \in \mathcal{R}$ is not literally a convex hull, but merely a list of points:

$$\sigma = \{u_1, \ldots, u_{k+1}\}. \tag{4}$$

In this case, a face is just a subset of $\sigma$:

$$\tau = \left\{u_{i_1}, \ldots, u_{i_{\ell+1}}\right\}. \tag{5}$$

Note that condition 3 (b) of Definition 1 is now vacuously true.

## B.2 HOMOLOGY GROUPS

Although we have not stated exactly how homology groups are defined, in this section we discuss a few of their properties. Fix a (possibly abstract) simplicial complex $\mathcal{S}$. We shall work with homology groups with coefficients in the field $\mathbb{Z}_2 := \mathbb{Z}/2\mathbb{Z}$, hence all homology groups will be vector spaces over $\mathbb{Z}_2$. We will focus on the zeroth and first homology groups, denoted $H_0(\mathcal{S})$ and $H_1(\mathcal{S})$ respectively. Elements in $H_0(\mathcal{S})$ are equivalence classes of points, where two points are equivalent if they are in the same path component of $\mathcal{S}$. Elements in $H_1(\mathcal{S})$ are equivalence classes of closed loops, where two loops are equivalent if they "encircle the same hole" (Munch, 2017). Consequently, the dimension of $H_0(\mathcal{S})$ (the zeroth *Betti number*, $B_0(\mathcal{S})$ counts the number of path connected components of $\mathcal{S}$, while the the dimension of $H_1(\mathcal{S})$ (the first *Betti number*, $B_1(\mathcal{S})$ counts the number of distinct holes in $\mathcal{S}$.

## B.3 THE RIPS COMPLEX

Given the above, in order to associate Betti numbers to a point cloud we first need to define an appropriate simplicial complex.

**Definition 2** (The Vietoris-Rips complex). Fix a point cloud $\mathcal{U} = \{u_1, \ldots, u_K\} \subset \mathbb{R}^n$, and for simplicity assume that $K < n$. Select a distance parameter $\epsilon \geq 0$. We define the simplicial complex $\mathcal{S}_\epsilon$, known as the Vietoris-Rips, or simply Rips, complex to contain all simplices

$$\sigma = \mathrm{Conv}\left\{u_{i_1}, \ldots, u_{i_{\ell+1}}\right\}, \tag{6}$$

satisfying the condition:

$$\max_{1 \leq m < n \leq \ell+1} \|u_{i_m} - u_{i_n}\|_2 \leq \epsilon. \tag{7}$$

In words, $\mathcal{S}_\epsilon$ contains all simplices on $\mathcal{U}$ with a diameter less than $\epsilon$. We note that $\mathcal{S}_0 = \mathcal{U}$ and hence $B_0(\mathcal{S}_0) = |\mathcal{U}|$, as every point in $\mathcal{U}$ is its own connected component., while $B_1(\mathcal{S}_0) = 0$. On the other end of the scale, when $\epsilon > \mathrm{diam}(\mathcal{U})$, where

$$\mathrm{diam}(\mathcal{U}) = \max_{1 \leq i < j \leq N} \|u_i - u_j\|_2, \tag{8}$$

the full-dimensional simplex

$$\sigma = \mathrm{Conv}\left\{u_1, \ldots, u_K\right\} \tag{9}$$

is contained in $\mathcal{S}_\epsilon$, and thus $\mathcal{S}_\epsilon$ has one connected component and no loops: $B_0(\mathcal{S}_\epsilon) = 1$, $B_1(\mathcal{S}_\epsilon) = 0$. Consequently, the important topological features of $\mathcal{U}$ are detected by the Rips complex for $\epsilon$ values in $(0, \mathrm{diam}(\mathcal{U}))$.

The condition $K < n$ in 2 may be removed, in which case $\mathcal{S}_\epsilon$ is defined as an abstract simplicial complex. This distinction is not relevant for our work.

### B.4 PERSISTENT BETTI NUMBERS

Which value of $\epsilon$ should one choose? As discussed in Munch (2017), the trick is not to select a particular value of $\epsilon$ but rather focus on features (concretely: equivalence classes in $H_0(\mathcal{S}_\epsilon)$ and $H_1(\mathcal{S}_\epsilon)$) which *persist* for large ranges of $\epsilon$. More specifically, we define $\epsilon_b$, the *birth time*, to be the value of $\epsilon$ at which a particular equivalence class is first detected in $\mathcal{S}_\epsilon$. The death time, $\epsilon_d$, is the largest value of $\epsilon$ for which a particular equivalence class is detected in $\mathcal{S}_\epsilon$. Fixing a threshold `thresh`, we say an equivalence class is *persistent* if $\epsilon_d - \epsilon_b > $ `thresh`. We define the persistent zeroth (respectively first) Betti number $B_0$ (respectively $B_1$) to be the number of distinct equivalence classes appearing in $H_0(\mathcal{S}_\epsilon)$ (respectively $H_1(\mathcal{S}_\epsilon)$) which satisfy $\epsilon_d - \epsilon_b > $ `thresh`.

### B.5 SLIDING WINDOW EMBEDDING

Finally, we note that Perea & Harer (2015); Tralie & Perea (2018) propose a more sophisticated method for detecting periodicity using the *sliding window embedding*, also known as the *delay embedding*:

$$\{u_j\}_{j=1}^K \mapsto \left\{ SW_{d,\tau}(u_j) := \begin{bmatrix} u_j \\ u_{j+\tau} \\ \vdots \\ u_{j+d\tau} \end{bmatrix} \right\}_{j=1}^{K-d\tau}, \tag{10}$$

where $\tau$ (the delay) and $d$ (the window size) are user-specified parameters. Then, persistent Betti numbers are computed for $\{SW_{d,\tau}(u_j)\}_{j=1}^{K-d\tau}$ instead. This construction is motivated by Taken's theorem Takens (2006) which, informally speaking, states that the dynamics of $\{u_j\}_{j=1}^K$ can be recovered completely from its sliding window embedding, for sufficiently large $d$. Note this comes at a price: the increase in dimension means an increase in computational cost.

In preliminary experiments we found that the persistent Betti numbers for $\{SW_{d,\tau}(u_j)\}_{j=1}^{K-d\tau}$ did not reveal anything that could not already be inferred from the persistent Betti numbers of $\{u_j\}_{j=1}^K$. Thus, we chose not to work with the sliding window embedding.

## C ADDITIONAL EXPERIMENTAL DETAILS

### C.1 ADDITIONAL DETAILS ON `PI-net`

During our experiments we determined that there was another model parameter, beyond the number of iterations, that had a strong impact on the accuracy of `PI-Net`. Specifically, there is a `threshold` parameter within the forward solver, Broyden's method, that controls the maximum rank of the inverse Jacobian approximation. Based on code included in the supplementary material for Anil et al. (2022), it appears the `threshold` parameter was originally set at 40. However, with this setting `PI-Net` performed very poorly; it failed on all mazes of size $49 \times 49$. Increasing `threshold` increased the accuracy of `PI-Net`, but also increases memory costs because it requires storing a number of high-dimensional latent iterates equal to `threshold`. For our experiments, we used `threshold = 1,000` in order to achieve strong accuracy without unreasonable memory requirements.

### C.2 COMPUTATIONAL RESOURCES

The experiments in this study were performed on a high-performance workstation with the following specifications:

- **CPU:** AMD Ryzen Threadripper PRO 3955WX (16 cores, 32 threads)
- **GPU:** NVIDIA RTX A6000 (48 GiB VRAM)
  - CUDA Version: 12.5, Driver Version: 555.42.06
- **Memory:** 251 GiB RAM
- **Operating System:** Ubuntu 20.04.6 LTS (x86_64 architecture)

## C.3 FAILED MODEL PREDICTIONS

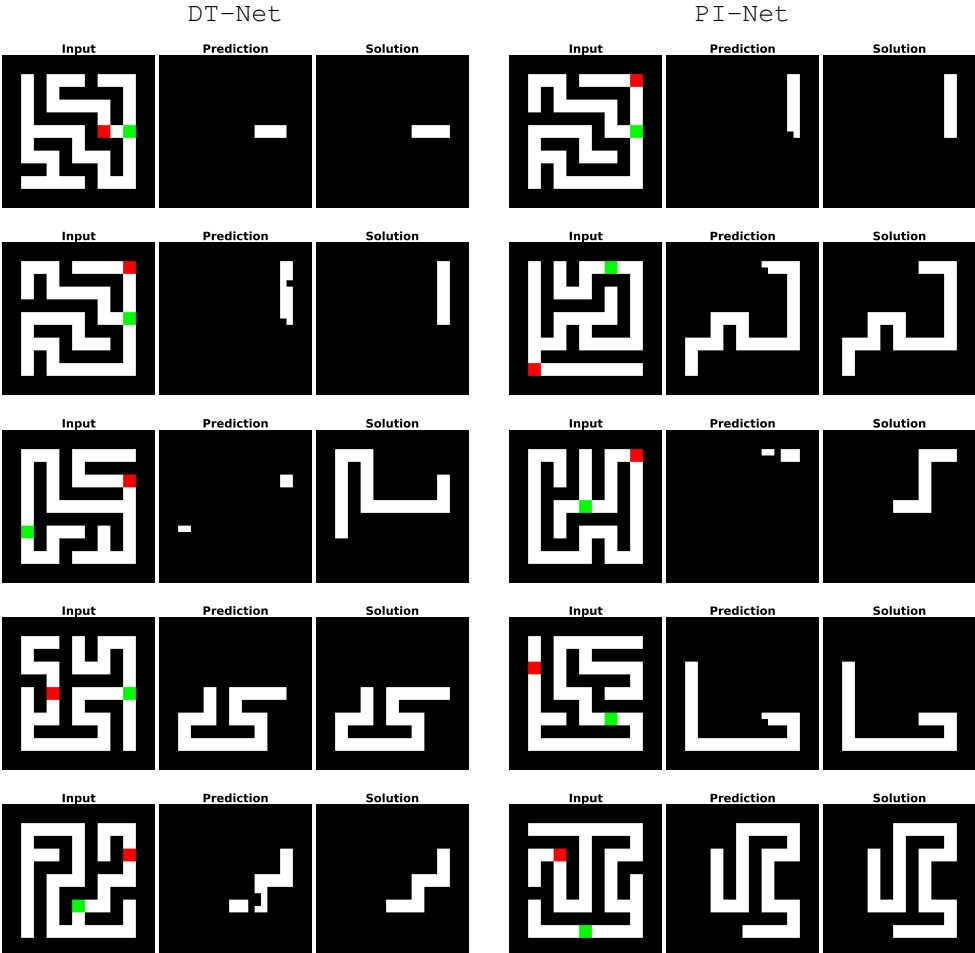

Figure 8: Examples of 9 × 9 mazes with deadend_start=False predictions from DT-Net (left) and PI-Net (right), some of which they fail to solve. Note that mistakes are often in the start position cell or cells immediately adjacent to it.

## D THE RIPSER WRAPPER

A number of optimizations were applied to reduce the memory and compute time of TDA. Instead of providing a sequence of high-dimensional latent iterates directly to Ripser, we use a smaller distance matrix containing pairwise distances between the iterates. This matrix is computed using an optimization from Tralie & Perea (2018), where the iterates are first compressed with singular value decomposition (SVD) to reduce memory costs. The SVD computation is performed in PyTorch to leverage GPU acceleration. Initially, we also used a diagonal convolution optimization, also from Tralie & Perea (2018), to avoid redundant computations when constructing the distance matrix for the sliding window embedding. However, this second optimization was not used in our final TDA experiments, as we dropped the sliding window embedding.

