# OpenReview forum: "On Logical Extrapolation for Mazes with Recurrent and Implicit Networks"
_ICLR.cc/2025/Conference — ICLR 2025 Conference Withdrawn Submission_

### Official Review · Reviewer_PzsY · 2024-10-27

**Soundness:** 2
**Presentation:** 2
**Contribution:** 2
**Rating:** 5
**Confidence:** 2

**Summary:**

This paper explores the logical extrapolation capabilities of recurrent neural networks (RNNs) and implicit neural networks (INNs) in the context of maze-solving tasks. The authors investigate whether these network architectures, when trained on simpler instances of a task (e.g., small mazes), can generalize to more complex instances (e.g., larger or differently structured mazes). Their main contributions include:

1. Demonstrating that while both RNNs and INNs can extrapolate to larger mazes, their generalization is less effective when maze complexity increases through factors other than size, such as changes in the starting point configuration or the introduction of cycles.

2. Introducing new difficulty axes, such as a deadend start and percolation parameters, which significantly impact model performance, revealing limits to previous assumptions about logical extrapolation.

3. Providing insights into the latent dynamics of RNNs and INNs through topological data analysis (TDA). The authors observe various limiting behaviors, such as convergence to fixed points or periodic cycles, and suggest these dynamics are essential for understanding the generalization potential of such networks.

Overall, the study underscores the need for careful consideration of difficulty dimensions in neural extrapolation tasks and suggests that topological tools may help improve understanding and robustness of extrapolative models.

**Strengths:**

1. Originality: The paper introduces new difficulty dimensions for maze-solving (e.g., deadend starts and maze percolation) beyond simple maze size, providing fresh insights into neural network generalization. The use of Topological Data Analysis (TDA) to explore latent space dynamics is also novel in this context and adds an interesting layer to understanding neural networks’ internal behaviors.

2. Quality: The experiments are well-structured, covering a variety of maze configurations. The analysis of latent dynamics using TDA methods, such as Betti numbers, is thorough and demonstrates different behaviors (e.g., fixed points, cycles), adding depth to the results.

3. Clarity: The paper is clear and well-organized, with effective visual aids that support understanding of key results. The explanation of TDA concepts is accessible, which is helpful for readers unfamiliar with this method.

4. Significance: This study highlights the limits of RNNs and INNs in extrapolating maze-solving tasks under varied conditions, offering valuable insights into neural network robustness and generalization. The methods and findings could inspire further research into network behaviors in complex problem-solving tasks.

**Weaknesses:**

1. Limited Applicability: The study focuses solely on maze-solving tasks, which may limit the generalizability of its findings. To strengthen the paper’s relevance, it would be beneficial to test RNNs and INNs on other logical extrapolation tasks or provide more justification for why maze-solving is a suitable proxy for broader reasoning abilities.

2. Complexity of TDA Analysis: The use of Topological Data Analysis (TDA) adds depth but may be challenging for readers not familiar with these tools. Including a simplified explanation or visual summary in the main text, or moving detailed technical aspects to an appendix, could improve accessibility and help readers better grasp the significance of TDA results.

3. Out-of-Distribution Performance: Although the study introduces new dimensions of difficulty (e.g., deadend starts, percolation), the models show limited ability to generalize effectively in these out-of-distribution scenarios. This indicates that the models may require further improvements or modifications. Exploring architectural adjustments or fine-tuning strategies to enhance generalization across these dimensions could improve the paper’s impact.

4. Interpretation of Latent Dynamics: While the paper presents examples of limiting behaviors (e.g., fixed points, cycles) in latent space, the connection between these dynamics and the models' generalization performance is somewhat limited. Providing a clearer theoretical analysis or practical insights on how these dynamics relate to extrapolation ability would make the findings more actionable and valuable.

5. Evaluation Depth: The evaluation could benefit from more in-depth comparisons with other types of neural networks beyond RNNs and INNs. Testing simpler architectures, such as CNNs or MLPs, on the same maze tasks would help contextualize the benefits and limitations of RNNs and INNs, strengthening claims about their suitability for logical extrapolation.

**Questions:**

1. Generalizability Beyond Maze-Solving: The study currently focuses on maze-solving as a test bed for logical extrapolation. How do the authors view the transferability of their findings to other logical reasoning or extrapolation tasks? Would they consider additional experiments or theoretical discussions on the applicability of their results beyond maze-solving?

2. Choice of TDA for Latent Dynamics Analysis: The use of Topological Data Analysis (TDA) provides unique insights, but could the authors clarify why TDA was chosen over other possible analysis techniques (e.g., standard trajectory or spectral analysis)? Additionally, have the authors considered other types of topological structures that may emerge in the latent space besides fixed points and cycles?

3. Relation Between Latent Dynamics and Model Performance: The paper presents different latent dynamics (e.g., fixed points, cycles), but the connection to model performance remains somewhat abstract. Could the authors elaborate on how these dynamics influence logical extrapolation? For example, are certain behaviors (e.g., two-loop cycles) generally associated with better or worse generalization?

4. Handling of Out-of-Distribution Shifts: The experiments show that both RNNs and INNs struggle with out-of-distribution shifts, such as deadend starts and percolation. Have the authors explored any methods (e.g., fine-tuning, architectural changes) to improve robustness to these shifts? Insights on potential adjustments would be valuable.

5. Explanation of TDA in the Main Text: TDA is an advanced method that may be unfamiliar to some readers. Would the authors consider adding a high-level explanation of TDA, perhaps with intuitive examples, to make the analysis more accessible? This could help readers better understand the significance of Betti numbers and their relation to latent dynamics.

6. Evaluation with Additional Network Architectures: The paper evaluates RNNs and INNs but does not compare them to simpler architectures. Could the authors include results for models like CNNs or MLPs on the same maze tasks? This would clarify how unique RNNs and INNs are in their extrapolation abilities and whether these findings are specific to recurrent or implicit architectures.

---

### Official Review · Reviewer_gSwi · 2024-10-28

**Soundness:** 3
**Presentation:** 2
**Contribution:** 1
**Rating:** 3
**Confidence:** 3

**Summary:**

This paper investigates "logical extrapolation" in previously published neural network models, specifically focusing on Recurrent Neural Networks (RNNs) and Implicit Neural Networks (INNs). Prior authors defined "logical extrapolation" as the model's capacity to add shared-weight layers during testing, enabling it to address more complex problems than those encountered during training. In this work, the authors tackle the challenge of connecting a start and end node within maze-solving tasks. Unlike previous studies that succeeded on extrapolating maze size, this paper demonstrates that top-performing models from prior research struggle to extrapolate when additional complexity arises across different dimensions. For example, slight distribution shifts—such as allowing multiple neighbors for start nodes, or introducing cycles into previously cycle-free mazes—impede the model's ability to extrapolate.

The authors conduct a straightforward yet insightful topological analysis of the network's latent space dynamics, illuminating the underlying behaviors and convergence patterns of these networks. Findings reveal that DT-NET, the highest-performing RNN model, does not converge to a single latent point, whereas PI_NET, the top-performing INN model, does. This dynamic is explored through principal component analysis, residual analysis, and homology grouping, uncovering a complex and partially unexplored latent behavior in DT-NET. Examining network dynamics may lead to more interpretable and robust extrapolation strategies utilizing recurrence.

The paper concludes that the extrapolation capabilities of DT_NET and PI_NET are significantly influenced by the specific axis of complexity introduced. These findings challenge previous assertions about general logical extrapolation and suggest further research into why certain extrapolation dimensions are more manageable for neural networks.

**Strengths:**

1. This work provides evidence that challenges the claims of previous research by setting up new tests where prior work fell short, underscoring the importance of establishing limitations on earlier findings.

2. The paper is well-structured across its five sections, presented with direct and concise wording, making it easy to read and follow.

**Weaknesses:**

1. The primary reason for rejection is that the paper does not solve or propose any new solutions; it merely highlights where previous work falls short, without suggesting a new model or method to address the identified issues. In my view, experiments without any novel proposals are insufficient for a conference of this caliber.

2. Testing on only 100 mazes (line 237) appears limited, especially given that prior work typically evaluates on a much larger sample size, often 1,000 to 10,000 mazes. I recommend that the authors consider increasing the test set size to provide more robust insights into model accuracy or provide a justification for why 100 mazes would be adequate for reliable performance assessment in comparison to these larger sample sizes.

3. Figure 7's legends are very small, making them difficult to read. I suggest increasing the font size of the legends to match or be close to the main font size of the paper for improved readability. Additionally, Figures 5 and 6 are not referenced in the text, and Figure 3.2 is misreferenced twice (lines 253 and 258), where Figures 5 and 6 should be referenced instead, please double check all figure references in the text.

4. Figure 5 suggests that only a few mazes were tested with four start neighbors, leading to an imbalanced distribution. I recommend testing more mazes with four start neighbors for a more balanced analysis or explaining why this case is less common or important. Additionally, the y-axis label “probability” is misleading, as it actually represents counts of correct and incorrect instances. A more accurate label, such as “Count” or “Number of Instances,” would improve clarity.

5. Line 355 references Table 4, although it should refer to Table 1, as this is the only table in the paper, please double check all references in the text.

**Questions:**

The paper would benefit from addressing the identified experimental issues with a proposed solution and performance evaluation, which would enhance the impact of the work. Additionally, careful correction of referencing errors throughout would improve the paper's overall presentation.

---

### Official Review · Reviewer_35PN · 2024-10-30

**Soundness:** 4
**Presentation:** 3
**Contribution:** 2
**Rating:** 3
**Confidence:** 3

**Summary:**

This paper studies RNNs and INNs on extrapolative maze-solving tasks. The authors show empirically that prior results on networks do not generalize when the task is made more difficult on a new axis. The authors also study the dynamics of the RNNs and INNs and show that the dynamics either converge to a fixed point or a cycle.

**Strengths:**

**Originality**

While the paper considers a task and architecture found in prior works, they conduct new analyses which reveal additional information about the inner mechanics of the model.

**Quality**

Overall, the experiments are thorough and well-conducted. The authors consider a number of ways to both extrapolate the task and analyze the latent dynamics.

**Clarity**

Overall, the paper is well-written and the figures are well-illustrated.

**Significance**

This paper will likely have some significance to researchers studying extrapolation on the specific maze task studied.

**Weaknesses:**

Overall, the paper seems quite similar to prior work on the maze extrapolation task, and the new analyses do not seem very significant. Certainly, the authors show new results, but the broader significance to extrapolative tasks is not clear. I would encourage the authors to consider at least one other extrapolative task. Another way to improve the significance of the paper is to include theoretical results.

Clarity-wise, some text in figures is too small (namely, 2, 3, 4, 5, 6, 7). I encourage the authors to increase figure font size throughout.

**Questions:**

What is the significance that the models tested cannot extrapolate well along the new axes of dead-end start and percolation? Can we apply this to other extrapolative tasks?

What is the significance of the convergence dynamics results on this task? Do similar patterns apply to other extrapolative tasks?

Can the authors show theoretical results on the extrapolative abilities of models on this maze task?

---

### Official Review · Reviewer_pBMP · 2024-10-30

**Soundness:** 3
**Presentation:** 3
**Contribution:** 2
**Rating:** 5
**Confidence:** 3

**Summary:**

The authors explores whether a recurrent neural networks (RNN) and a implicit neural networks (INN) exhibits logical extrapolation, finding they do with extrapolate well respect to maze size, but extrapolate less well with respect to dead-end-start and percolation (loop paths). PCA analysis of latents shows 3 distinct modes. Topological Data Analysis (TDA) supports this finding

**Strengths:**

This is an interesting paper.

Good analysis of two models in 3 extrapolation dimensions.

Good mathematical analysis of the latents of the models.

Content is well presented.

**Weaknesses:**

The paper would benefit from going deeper in a few areas. Refer Questions section

The PCA/TDA finding of one point, two point, two circles is very interesting. Understanding how these modes relate to model performance, algorithm or similar would extend this finding. Without some implications of this finding, it’s hard to say how important this finding is.

Erata:
- Page 5. Text “mazes still satisfying this condition contribute” is ambiguous. Consider using “mazes with a start position degree of 1 contribute” if this is the meaning
- Page 5. “See Figure 3.2 for a breakdown of accuracy by start position” seems incorrect. There are multiple references to this figure that seem incorrect.
- Inconsistent capitalization of DT-Net as DT-net. Ditto PI-Net.
- Page 7. “Table 4” links to Table 1. Check all links.

**Questions:**

More comparison of the model success in different extrapolation directions would be useful. Consider discussing the models in terms of the features the base models have vs what features the different extrapolation directions require. Perhaps:
- Increasing maze size is “more of the same” with the model able to reuse existing features.
- Deadend start = false increases the search space size, eliminating a few initial “forced” moves. A deeper/longer solution path, increases the compute required. Figure 5 reflects this.
- Base models were trained with percolation = 0, and so learnt to find the unique solution. They were not trained to select one path out of several valid paths. So Figure 6 results seem unsurprising. To pick one out of several valid paths (e.g. the shortest) the model would need a feature that distinguishes between valid paths (e.g. measure path length, and prefer minimum path length). To learn this feature likely requires more model training.

---

### Official Review · Reviewer_XdCP · 2024-11-03

**Soundness:** 3
**Presentation:** 3
**Contribution:** 1
**Rating:** 3
**Confidence:** 3

**Summary:**

This paper analyzes the behavior of maze-solving RNNs and INNs. They take models from prior work and evaluate their generalization performance on a dataset that consists of maze problems that are harder than these models’ original training data. Unlike the training data, the test data contains mazes of larger sizes, mazes where the start node is not a dead end, and mazes with cycles. They find that the models studied generalize to larger mazes but not in the latter categories. They also use topological data analysis to study the behavior of the latent data after more RNN/INN iterations.

**Strengths:**

1. The use of topological data analysis is to study sequences of RNN latents is novel, to my best knowledge.
2. The description of the experimental setup is clear and the main points in the paper are conveyed in a way that is easy to understand.

**Weaknesses:**

I believe that the depth of analysis does not meet the bar of an ICLR acceptance.

This paper can be seen as a paper about generalization or interpretability. When evaluated as a paper about generalization:

1. The paper observes a lack of generalization but does not provide much surprising insight or propose new techniques to improve generalization. I believe the Bansal work already covers the claim about their model generalizing to increased maze size, so the new claims are about deadend-starts and cycles. The explanation for this lack of generalization is simple:

“Both DT-net and PI-net are trained to find the unique path from start to end, but when the maze has even a single loop, there  is no longer a unique path. When presented with a maze that does not have a unique solution path, both models fail (see Figure 3.2 and Appendix Subsection C.3), whereas a human might reasonably reinterpret the task (e.g. “find the shortest path”, or even “find a path”) and solve it.”

The problem setup changes so certain invariants are no longer respected and the problem may no longer be well-defined. That models can fail to generalize under these conditions seems like an intuitive and well-known fact, e.g. https://arxiv.org/abs/2210.01790, https://arxiv.org/abs/2105.14111.

2. The lack of generalization seems specific to the models studied. The authors do not explore whether we can easily correct for this, for instance by training on a small sample of hard data.

3. The maze dataset itself is not new, so the extent of the analysis is to run previously trained models on an open source dataset.

4. Failure modes are not sufficiently explored.

When evaluated as a paper about interpretability, the analysis is observational, mostly making statements about what the latents look like and doesn’t try to explain why the dynamics occur or what the mechanism is for the model to arrive at periodic latents. It’s well known that latents may only be spuriously correlated with model mechanisms and may not have much meaning on their own, which is why most interpretability work includes causal evidence. For example, it’s possible that the latents are periodic in a direction that isn’t read by the final projection matrix, in which case this finding isn’t very interesting. TDA seems like an intriguing direction but needs to be supplemented with further analysis to see whether it relates to the models’ lack of generalization (though it seems somewhat unrelated, given that the PI-net always converges to a fixed point yet still fails to generalize).

**Questions:**

In appendix C.3, it looks like predictions sometimes fail due to lack of exact match between prediction and solution by a few pixels, but not by a whole maze “square”. Does this failure mode occur often in the training data, and how much does this explain additional failures in test?

---

### Official Review · Reviewer_wsym · 2024-11-04

**Soundness:** 3
**Presentation:** 3
**Contribution:** 3
**Rating:** 6
**Confidence:** 4

**Summary:**

In this paper, the authors question the ability of recurrent and implicit neural networks in logical extrapolation. The authors demonstrate that prior approaches to logical extrapolation using RNNs and INNs on maze solving fail when the difficulty scales along dimensions other than maze size (degree of the start of mazes, presence of loops in the maze). Additionally, the authors also present an analysis of the RNN/INN latent dynamics where they highlight key signatures of either model's trajectory in the hidden dimension space.

**Strengths:**

* Logical extrapolation is a key ability of human vision. The authors show that prior work in this area need more exploration and highlight that scaling difficulty in dimensions other than model size hurts the out-of-distribution performance of RNNs and INNs, both of which have been the kind of networks which have shown logical extrapolation. This is a really interesting contribution and warrants the need to further explore this unsolved problem.
* This work includes source code for generating the various difficulty levels of mazes (deadend_start and percolation), this is great and promotes tackling a more holistic solution to logical extrapolation by the community.
* The writing, figures and accompanying captions have sufficient detail and the paper is quite a straightforward read.

**Weaknesses:**

* **Loose connection between the TDA analysis and lack of logical extrapolation**: It felt to me that the TDA analysis of RNN/INN latent dynamics is disconnected from the issue of logical extrapolation in RNNs and INNs. My biggest feedback to improve this paper further would be to strengthen the connection between these two explorations in the paper's writing as I believe this would greatly help in gleaning the contributions clearly.
* **Overall framing of the story**: The current writing sounds quite critical of RNN/INNs ability to perform logical extrapolation. These are currently the main class of models from prior art that show (any) extrapolation to greater difficulty settings, and I felt that the writing was potentially overly critical of these model architectures. This is a minor weakness in my opinion, yet I still find it concerning since we don't have another architecture class of neural networks that do a better job at extrapolating to out of distribution settings in solving mazes.

**Questions:**

* Have the authors tried reframing the problem to classification instead of segmentation? Say, the authors were to generate a maze classification dataset where the stimuli were either mazes with a solution and mazes without a solution (where the green and red dots are disconnected), would generalization still be hurt in the other dimensions evaluated? Since the authors are only basing their conclusions in one dataset (Maze solving), the presented results may be more compelling when discussed with converging evidence on both settings of the problem.

---

### Note · Authors · 2024-11-25

**Comment:**

We thank the reviewers for their time and constructive feedback. We are grateful that our work was received as "well presented" (reviewer pBMP), "well structured and easy to read" (reviewer gSwi) and that the novelty of our use of topological data analysis was noted (reviewers XdCP, pBMP, PzsY).

We also thank the reviewers for suggesting ways in which our paper could be bolstered by, for example, additional experiments. In order to maintain the quality and clarity of our work, we have decided to **withdraw our paper** while we consider which of these additional directions to pursue.

**Withdrawal Confirmation:**

I have read and agree with the venue's withdrawal policy on behalf of myself and my co-authors.